# Seasonal and Interannual Variability of the Indo-Pacific Warm Pool and its Associated Climate Factors Based on Remote Sensing

**Zi Yin** [1,2] , **Qing Dong** [1,*], **Fanping Kong** [1,2], **Dan Cao** [1,2] **and Shuang Long** [1,2]

[1] Key Laboratory of Digital Earth Science, Aerospace Information Research Institute, Chinese Academy of Sciences, Beijing 100094, China; yinzi@radi.ac.cn (Z.Y.); kongfp@radi.ac.cn (F.K.); caodan@radi.ac.cn (D.C.); longshuang@aircas.ac.cn (S.L.)
[2] University of Chinese Academy of Sciences, Beijing 100049, China
[*] Correspondence: qdong@radi.ac.cn; Tel.: +86-010-8217-8121

**Abstract:** With satellite observed Sea Surface Temperature (SST) accumulated for multiple decades, multi-time scale variabilities of the Indo-Pacific Warm Pool are examined and contrasted in this study by separating it into the Indian Ocean sector and the Pacific Ocean sector. Surface size, zonal center, meridional center, maximum SST and mean SST as the practical warm pool properties are chosen to investigate the warm pool variations for the period 1982–2018. On the seasonal time scale, the oscillation of the Indian Warm Pool is found much more vigorous than the Pacific Warm Pool on size and intensity, yet the interannual variabilities of the Indian Warm Pool and the Pacific Warm Pool are comparable. The Indian Warm Pool has weak interannual variations (3–5 years) and the Pacific Warm Pool has mighty interdecadal variations. The size, zonal movement and mean SST of the Indian Ocean Warm Pool (IW) are more accurate to depict the seasonal variability, and for the Pacific Ocean Warm Pool (PW), the size, zonal and meridional movements and maximum SST are more suitable. On the interannual scale, except for the meridional movements, all the other properties of the same basin have similar interannual variation signals. Following the correlation analysis, it turns out that the Indian Ocean basin-wide index (IOBW) is the most important contributor to the variabilities of both sectors. Lead-lag correlation results show that variation of the Pacific Ocean Warm Pool leads the IOBW and variation of the Indian Ocean Warm Pool is synchronous with the IOBW. This indicates that both sectors of the Indo-Pacific Warm Pool are significantly correlated with basin-wide warming or cooling.

**Keywords:** remote sensing; Indo-Pacific warm pool; El Niño southern oscillation; Indian monsoon; multi-time scale variability

## 1. Introduction

The warm pool is normally defined as the enclosed ocean area by an isotherm of a certain sea surface temperature (SST) in the range of 27.5–29 °C [1–3]. The warm pool regions of the tropical Indian Ocean and the western tropical Pacific Ocean constitute the Indo-Pacific Warm Pool [4], which contains a vast volume of the warmest ocean water and is a major source of heat and water vapor on earth [5]. Exceeding the threshold temperature of the atmospheric deep convection, the Indo-Pacific Warm Pool could therefore influence the global climate in many ways, such as influences on the Hadley circulation and Walker circulation or effects on the East Asian monsoon and the tropical cyclones [6–12].

Previous studies have focused on investigations of the warm pool variabilities on multiple time scales and the related atmospheric and oceanic processes [13–18]. For instance, surface heat flux,

SST, 20 °C isotherm depth, upper ocean heat content, sea surface salinity, warm pool centroids and edges and many other properties of the warm pool were constantly studied [3,4,19–21]. Through detecting oscillations of the eastern edge positions of the western Pacific warm pool (WPWP), it turns out that the South Warm Tongue and the North Warm Tongue were controlled by the annual cycles and were slightly affected by the El Niño onsets, whereas the East Cold Tongue was totally controlled by the El Niño onsets and nearly had no obvious annual cycle [20]. Not only the thermal edges of the warm pool were investigated but also the chlorophyll-a front in the eastern of the WPWP was detected by the satellite-based ocean color data. It was found that the eastern edge of the oligotrophic waters was characterized by a sharp transition of the chlorophyll-a concentrations and the zonal displacements of the edge significantly correlated to the displacements of the convergence zone of equatorial currents [21]. Lots of researchers have revealed variabilities of the warm pool center, which is an evident feature of the warm pool. Based on the satellite observation, Ho et al. [20] discovered that the centroid of the WPWP moved clockwise in the El Niño years but counterclockwise in the La Nina years during 1982–1991. Hu and Hu [22] defined the three-dimensional center of the WPWP and its seasonal variabilities were found mainly in the meridional and vertical directions, and interannual variabilities were found in the zonal and vertical directions. In addition, the zonal variation was found leading the Niño 3 index 3-4 months and the depth variation was found lagging the index for three months. The time-frequency features of the central latitude of the WPWP have also been investigated on multi-time scales from 1948 to 2012 in detail [23]. Moreover, Kim et al. [4] firstly introduced five warm pool surface properties to delineate the variabilities of the warm pool and contrasted the similarities and differences of the Indian Ocean sector and the Pacific Ocean sector on the seasonal and interannual time scale.

Various studies of the linkage of the warm pool with large scale atmospheric circulation and oceanic processes have been launched [24–27]. The upper-layer volume fluctuations of the WPWP were found closely related to the oceanic circulation driven by the trade wind, and during El Niño events, the eastern sector volume increased yet the western sector decreased in much less magnitude [20]. The subsurface ocean temperature in the WPWP always had continual positive (negative) anomalies in advance of El Niño (La Nina), which would propagate eastward and directly led to the outbreak of the El Niño (La Nina) [28]. The atmospheric processes were found to have competing effects on the warm pool's variability, the heat transport tended to homogenize SST while the atmospheric humidity and surface wind tended to remove homogeneity [29]. Meridionally symmetric precipitation and wind anomalies were observed over the central Pacific during the decaying phase of the Central Pacific (CP) El Niño events [30]. The warm pool had evident effects on the summer monsoon in East Asia: when the SSTA in the warm pool was positive (negative), the summer monsoon would be stronger (weaker). In addition, the positive (negative) SSTA in the warm pool was related to the East Asia-Pacific-America (EPA) wave-train stretching northwards to higher (lower) latitudes [28].

Inspired by the above research, we speculate that the Indian Ocean sector and the Pacific Ocean sector of the Indo-Pacific Warm Pool (IPWP) would have coincident and different variabilities on multi-time scales, which may correlate with large-scale climate modes. Therefore, we systematically analyze the differences and similarities of the two warm pool sectors' variabilities based on long time series remote sensing data, their time-frequency variation patterns as well as the associations with different climate modes and possible mechanisms of the variabilities.

The rest part of this paper is organized as follows. Section 2 introduces the relevant data and analysis methods. The trend of the warm pool properties in the IPWP is addressed in Section 3, as well as the seasonal and interannual variabilities and the associated climate patterns. Discussion remarks and the main conclusions are presented in Sections 4 and 5.

## 2. Materials and Methods

### 2.1. Data

To delineate the warm pool and to extract the properties, a satellite-measured daily mean SST dataset is used. The SST dataset with a high resolution of 0.25° by 0.25° is provided by the NOAA/OAR/ESRL PSD, Boulder, Colorado, USA, which can be obtained from https://www.esrl.noaa.gov/psd/. The NOAA 0.25° daily Optimum Interpolation Sea Surface Temperature (OISST) is constructed by combining observations from different platforms (satellites, ships, buoys) on a regular global grid, and its time coverage is from September 1981 to now. We use the daily OISST to resample a monthly mean SST dataset for our investigation.

The Eastern Pacific (EP) ENSO is represented by the Nino 3.4 index, which is calculated by the mean SST in the region over 5°S-5°N, 170°-120°W. The Central Pacific (CP) ENSO is identified by the El Niño Modoki Index (EMI) (Ashok et al.; 2007), the equation is EMI = $[SSTA]_C - 0.5[SSTA]_E - 0.5[SSTA]_W$, where the [SSTA] with different subscripts represent the average SST anomalies (SSTA) in the area over the central Pacific region C (10°S-10°N, 165°E-140°W), the eastern Pacific region E (15°S-5°N, 110°-70°W) and the western Pacific region W (10°S-20°N, 125°-145°E), respectively. The IOD index is defined by Saji et al. [31], which is the area-average SST over the Western Indian Ocean (10°S-10°N, 50°-70°E) minus the area-average SST over the Eastern Indian Ocean (10°S-0°, 90°-110°E). The Indian Ocean basin-wide (IOBW) index, following Zhou et al. [23], were defined as the area-average SST over the tropical Indian Ocean (20°S-20°N, 40°-100°E).

To investigate the variability of the warm pool sectors belonging to different oceans, we divided the warm pool into two demarcations based on the basin mask provided by NOAA's National Oceanographic Data Center [32]. The boundary is from the southern tip of the Indo-China Peninsula, through the Indonesian island chain to the northern tip of Australia. Hereafter, the two sectors of the IPWP will be called the Indian Ocean Warm Pool (IW) and the Pacific Ocean Warm Pool (PW), respectively.

In this study, all the data from 1 January 1982 to 31 December 2018 are used.

### 2.2. Methods

We define the IPWP as the region enclosed by the 28°C isotherm over the tropical Indian Ocean and the western tropical Pacific Ocean (30°S-30°N, 30°E-130°W), following Kim et al. [4]. The surface size, maximum and mean SSTs of the IPWP and both sectors are calculated to represent the intensity of the warm pool. The surface size is calculated as the sum of the grids of the identified warm pool (whether it is the IPWP or one of the demarcations). The area-averaged value and the maximum value of the identified warm pool's SSTs are the mean SST and the maximum SST, respectively. The zonal center and the meridional center are calculated using the following equations, which were improved by Hu and Hu [22]:

$$C_z = \sum_{i=1}^{n} (w_i \times x_i), \tag{1}$$

$$C_M = \sum_{i=1}^{n} (w_i \times y_i), \tag{2}$$

$$w_i = \frac{T_i - T_{min}}{\sum_{j=1}^{n} (T_j - T_{min})}, \tag{3}$$

where $x_i$ and $y_i$ are the grid i's longitude and latitude, respectively. The total amount of grids of the objective warm pool is the n, and $w_i$ is the weighting for the grid i in the center's calculation, which is calculated by Equation (3). In Equation (3), $T_i$ is the temperature of the grid i, $T_{min}$ is the minimum SST in all of the warm pool grids.

Trend testing utilizes the Mann-Kendal method, seasonal variation period is detected by the harmonic analysis, and the time-frequency patterns are found by the wavelet transform method. All the correlation coefficients are tested under Pearson's value.

## 3. Results

### 3.1. The SST Climatology and Trend of the Indo-Pacific Warm Pool

To have an overview of the IPWP, firstly we examine the long-term trend of the IPWP on its size and intensity using the Mann-Kendall method. Both time series of the size and the mean SST of the IPWP exhibit increasing trends during 1982–2018 (Figure 1), which are statistically significant above the 95% level. The size's increasing rate is 14.24 grids/year (approximate $1.53 \times 10^5$ km$^2$/year); the mean SST's increasing rate is $2.41 \times 10^{-4}$ °C/year.

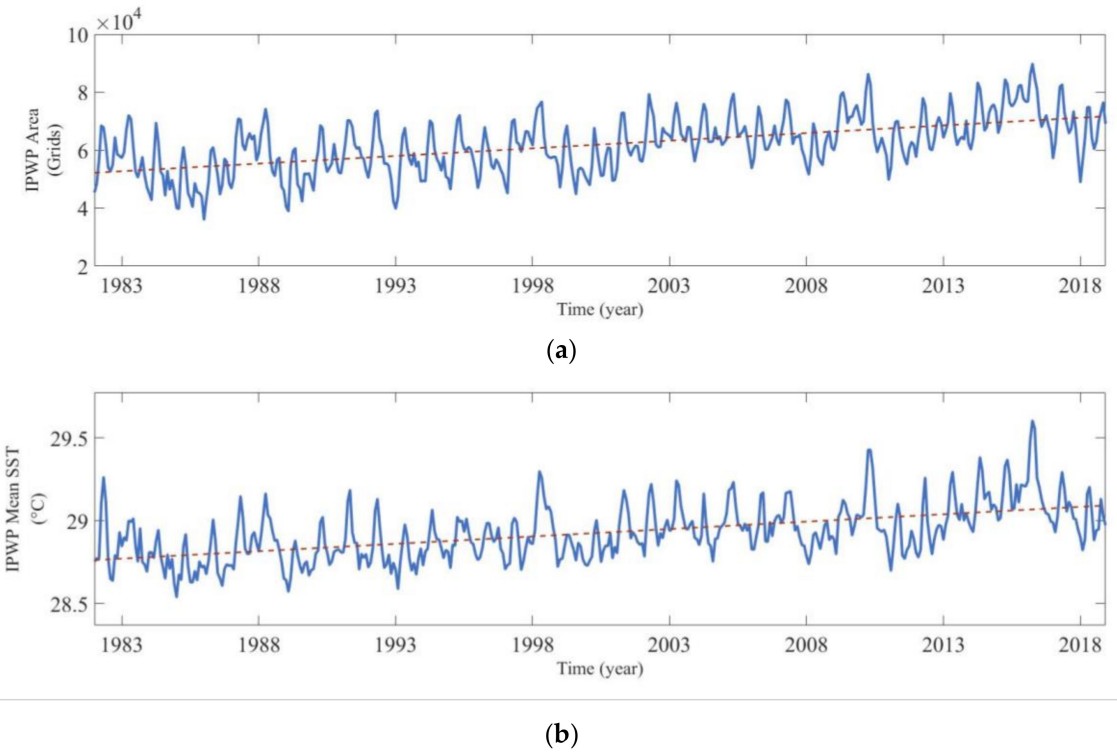

(a)

(b)

**Figure 1.** Time series of the Indo-Pacific Warm Pool (IPWP) size (**a**) and of the IPWP mean Sea Surface Temperature (SST) (**b**) with their trend line (red dotted line) superimposed.

Then the monthly climatology SST of the IPWP over 37 years (1982-2018) is illustrated in Figure 2. The surface size and intensity of the IPWP vary remarkably during seasonal evolution. The basin line divides the IPWP into the Indian Ocean sector and the Pacific Ocean sector, and they are visually different in variations. The most prominent seasonal variation in the PW is the meridional movement, whereas its size or intensity variation is not apparent. As for the IW, the size, intensity and movements change vigorously during the seasonal evolution.

The two sectors' proportions in the IPWP during the seasonal evolution are shown in Figure 3. The Pacific sector is consistently bigger than a half, and it reaches the largest proportion in August and hits the least proportion in March. Furthermore, the Indian sector is vice versa. Particularly in boreal spring, the IW expands to cover most of the northern Indian Ocean. Meanwhile, the proportion of the IW in the IPWP is the topmost.

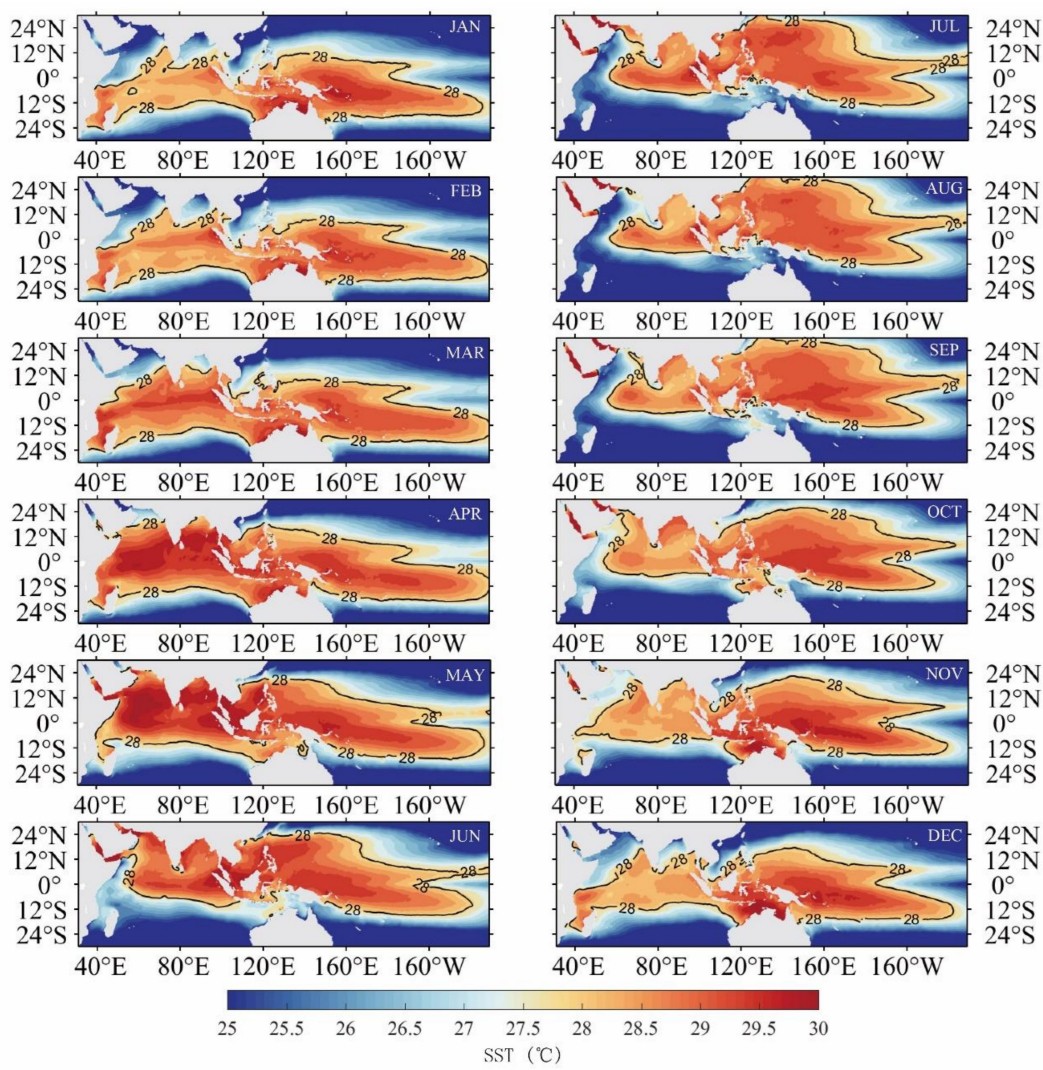

**Figure 2.** SST monthly climatology of the IPWP.

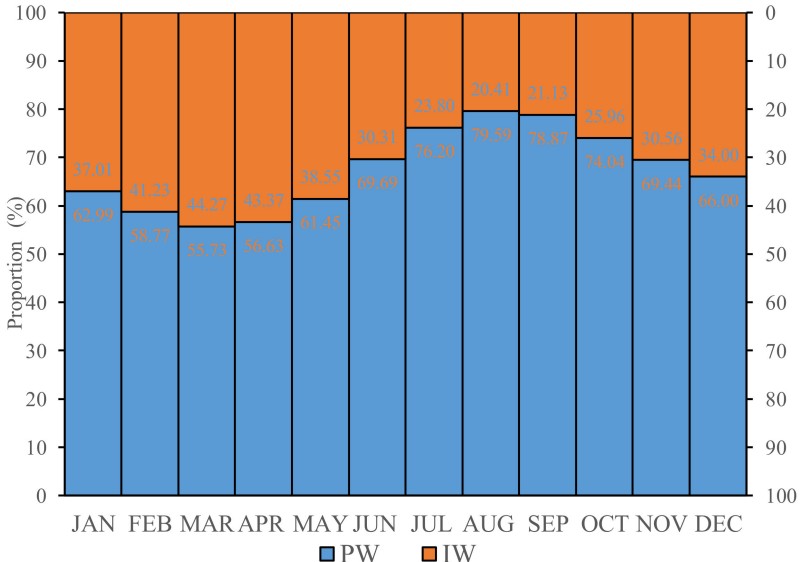

**Figure 3.** The proportions of the Indian Ocean Warm Pool (IW) and Pacific Ocean Warm Pool (PW) in the IPWP during seasonal evolution.

The long-term trend shown in Figure 1 and the seasonal variations shown in Figures 2 and 3 suggest that the IPWP becomes warmer which coincides with the global warming trend, and in consequence, it is expanding, and the seasonal variation is possibly controlled by the annual march of the solar heating.

*3.2. Multi-time scale Variabilities of the Indo-Pacific Warm Pool in Separated Demarcations*

3.2.1. Seasonal Variabilities of the Indo-Pacific Warm Pool in Separated Demarcations

Five warm pool properties including the size, zonal and meridional center, maximum and mean SST for the IPWP, IW and PW are calculated using the monthly SST dataset. Their seasonal anomaly from respective long-term means (left panels) and variation periods detected by the harmonic analysis (right panels) are shown in Figure 4. The IW size varies dramatically during seasonal evolution and is dominated by an annual harmonic (Figure 4a). The IW reaches its biggest size in April then keeps shrinking until August. Although the PW size is also dominated by an annual harmonic, the variation phase is almost opposite, the PW expands to its largest in September and then becomes smaller until February. Due to the IW's great size variation, the IPWP size change is consistent with the IW, though in a much lower magnitude. Meanwhile, the IPWP size also has a certain bimodal structure and is dominated by a semiannual harmonic, influenced by the opposite seasonal variation trends of the IW and the PW.

The zonal center of the IW has approximately weak harmonics of annual, semiannual and triple-annual (Figure 4b), as the zonal moving direction reverses several times during the seasonal cycle. The IW moves westward during boreal winter and spring, while it moves eastward during boreal summer and autumn. However, there is a slight fallback in September, which leads to the weak semiannual harmonic signal. As for the PW, it is dominated by an annual harmonic, where it moves westward during boreal spring and early summer, then shifts eastward from late summer to the end of winter. The zonal movement characters of the IW and PW seem to be primarily connected to the Indian Monsoon (changes in spring and autumn) and east Asian Monsoon (changes in summer and winter), respectively. The meridional movement variations of the IW and PW are fairly similar, with the same annual harmonic dominated and comparable variation magnitude (Figure 4c). They both move northward during boreal spring and summer and they move southward during boreal autumn and winter. However, there is still a difference between them, as the IW's meridional movement has a weak semiannual harmonic. All the same, we can also owe the seasonal meridional movement variation to the annual march of solar heating.

In terms of the seasonal variations of the warm pool intensity, it appears that the PW is more stable than that of the IW (Figure 4d,e). The seasonal variabilities of the maximum SST and mean SST of the PW are relatively small with a weak semiannual harmonic dominated. The maximum SST meets its peak twice in May and December, respectively, as the December peak is the higher one. Accordingly, it has two valley values in September and April, with the lower value in September. Similarly, the mean SST reaches its maximum in May and November, whereas it minimizes in March and August. The intensity of the IW varies much more vigorously throughout the year. The maximum SST is dominated by a semiannual harmonic and has a bimodal structure with peaks in July and December and valleys in October and February. The IW maximum and minimum values occur in July and October, which is different from the PW. Uniquely, the mean SST of the IW is dominated by an annual harmonic, which reaches its highest in May and lowest in November.

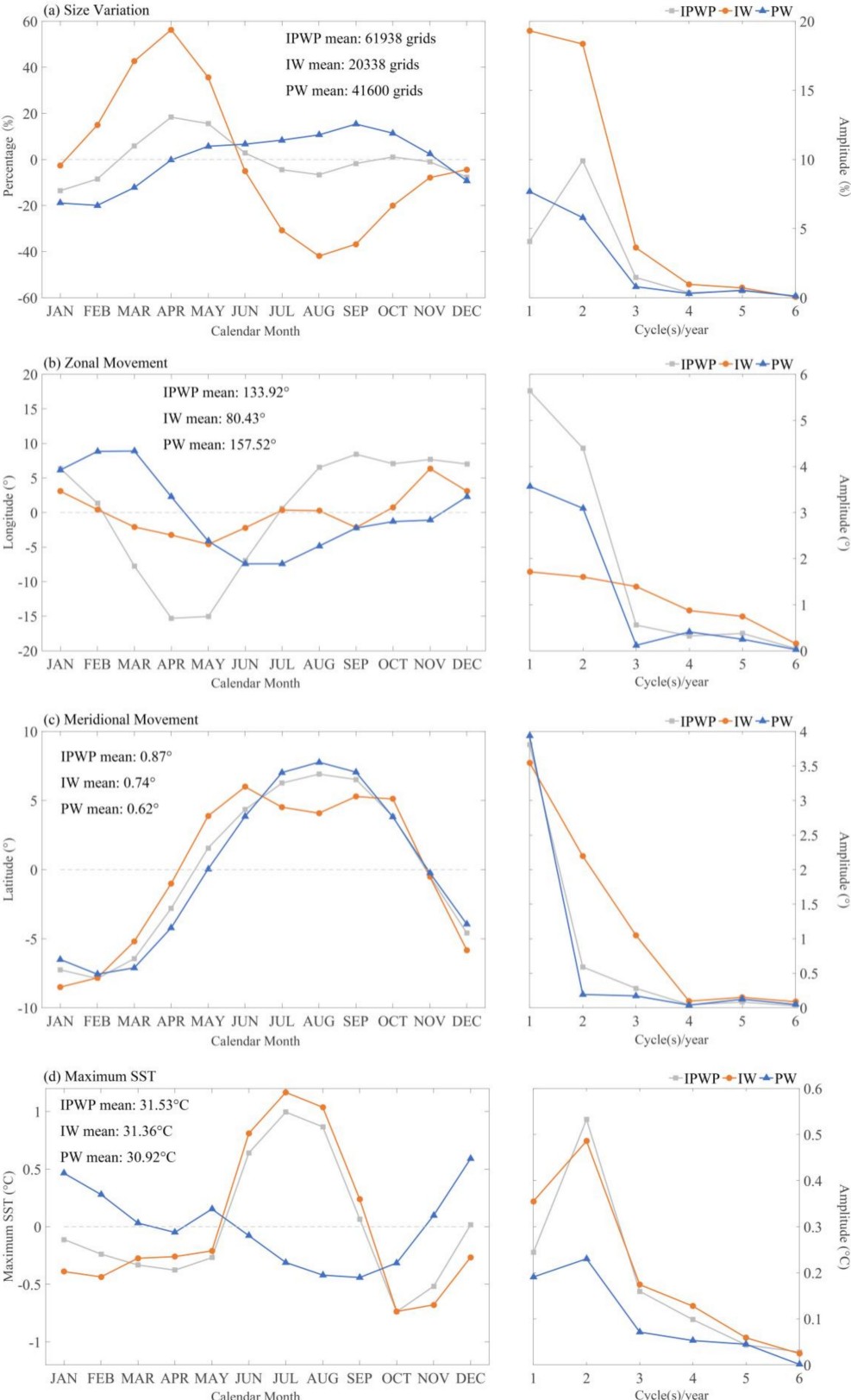

**Figure 4.** *Cont.*

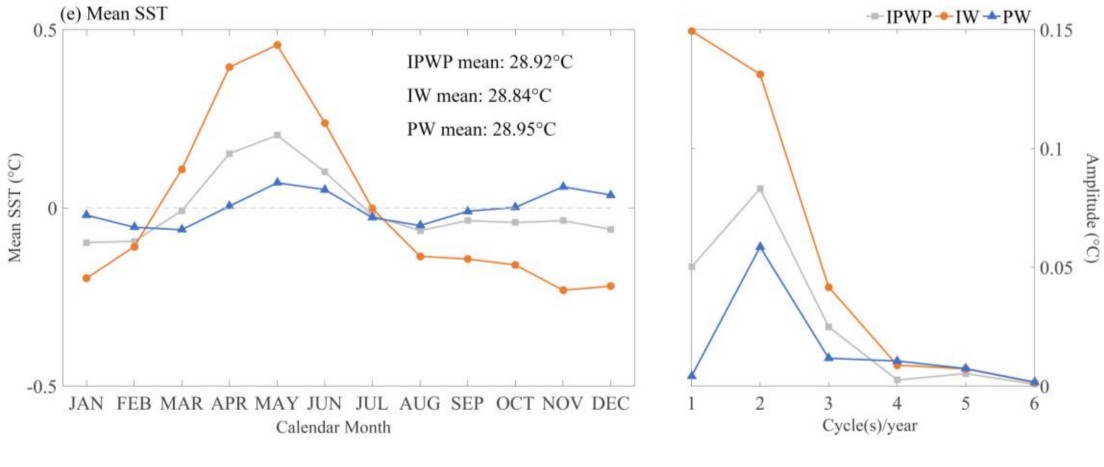

**Figure 4.** Annual cycles and harmonics of (**a**) Size variation; (**b**) Zonal center movement; (**c**) Meridional center movement; (**d**) Maximum SST; (**e**) Mean SST of the IPWP, IW and PW, respectively.

To specifically understand the seasonal evolutions of the warm pool properties between the demarcations and between themselves, we normalized the seasonal series of the properties (Figure 5, for the convenience of identification, west is the positive direction in this figure) and analyzed the correlations.

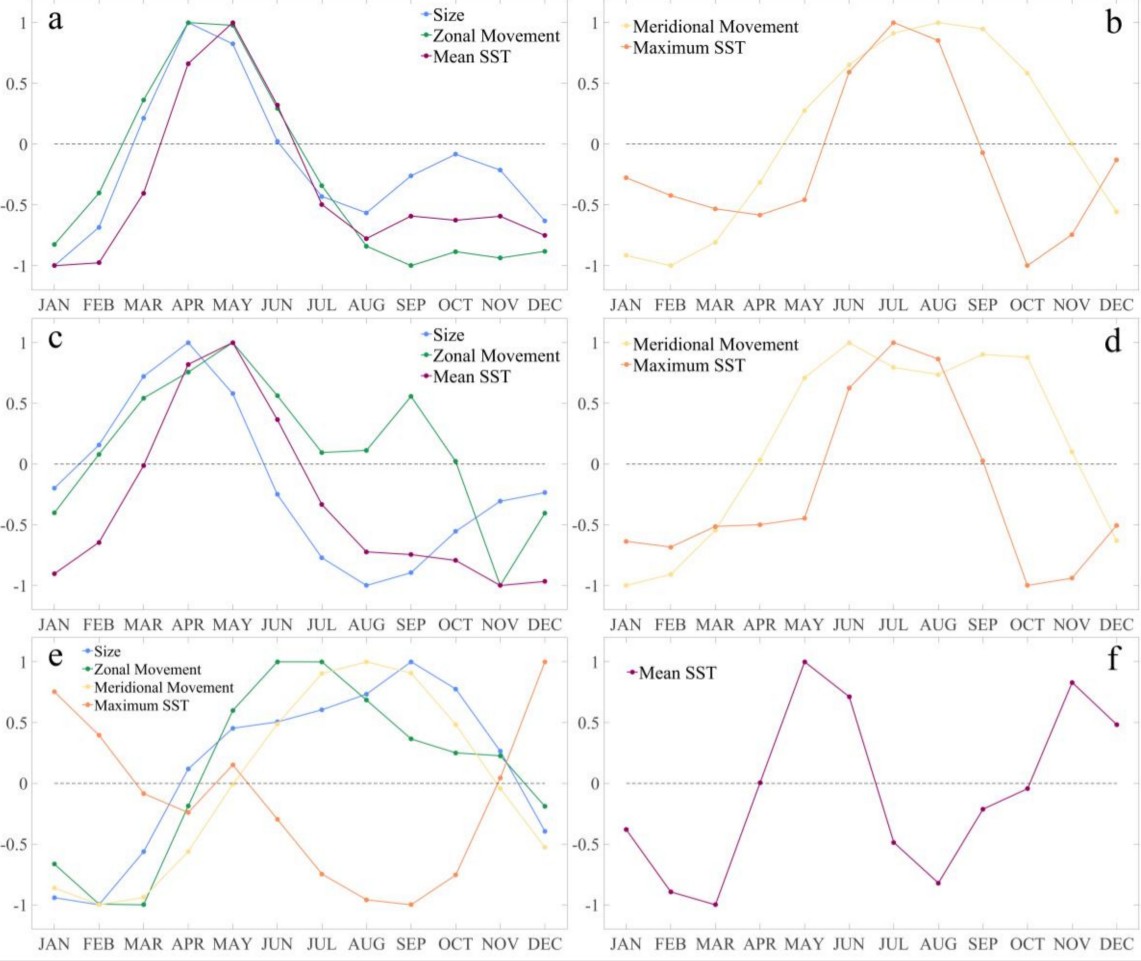

**Figure 5.** Annual cycles of normalized (**a**, **b**) IPWP properties; (**c**, **d**) IW properties; (**e**, **f**) PW properties.

The IPWP mean SST is highly correlated with the size (R = 0.91) and the zonal movement (R = −0.88). According to Figure 5a, during the seasonal evolution, the IPWP size expands (shrinks) simultaneously with the zonal center moves westward (eastward) and the mean SST increase (decrease) followed. The IW mean SST is significantly correlated with the size (R = 0.68) and the zonal movement (R = −0.81). According to Figure 5c, increase (decrease) of the IW mean SST follows the expansion of the IW size and the westward (eastward) movement of the zonal center, and increase (decrease) of the mean SST followed. The PW size is highly correlated with the zonal movement (R = −0.86), meridional movement (R = 0.92) and the maximum SST (R = −0.82). According to Figure 5e, during the seasonal evolution, the PW size always expands (shrinks) simultaneously with the zonal center moves westward (eastward), the meridional center moves southward (northward) and the maximum SST decreases (increases).

The size and the mean SST are more likely to explain the interseasonal variability of the IW, which is that the IW reaches its strongest status in boreal spring and the weakest status in autumn. The north boundary of the Indian Ocean is uniquely constituted by the Asian continents and therefore the IW is cornered to its smallest status in the northeast in late boreal summer (can also be seen in Figure 2). On the other hand, the PW status is relatively stable throughout the year. Although the PW gets to the largest size in boreal summer by spreading to the westmost and northmost, the PW mean SST does not vary much and seems to be controlled by the variation of solar radiation, i.e.; when the solar radiation marches away from the equator to the tropic of Cancer or the tropic Capricorn, the mean SST increases, and vice versa.

All the seasonal meridional movements strictly follow the sun's march, which could not be an ideal property to explain the seasonal variability of the warm pool. Curiously, the maximum SSTs of the IW and the IPWP are not relevant to the other properties, but the PW maximum SST is negatively correlated with the other properties of itself, except for the meridional movement. To investigate the underlying reason, we illustrate the positions of the seasonal maximum SST in Figure 6. It is obvious that the maximum SSTs are mostly located off the coast. With the movement of the IW, the maximum SSTs either locate off the north coast of Australia or in Madagascar strait when the IW moved to the south side, or the maximum SSTs locate in the Gulf of Oman in summer. In general, on account of the circumjacent continents, the IW maximum SST is not significantly correlated with the IW variability itself. On the other hand, we already know the maximum SST of the PW is negatively related to the other properties shown in Figure 5e. Figure 6 shows that the PW maximum SSTs moves from the north coast of Australia, then to the Maritime Continent, then to the northwest Pacific. Together with the movement, the maximum SST decreases. In other words, while the PW is expanding to the north and the west, without continents' effect the maximum SST is decreasing.

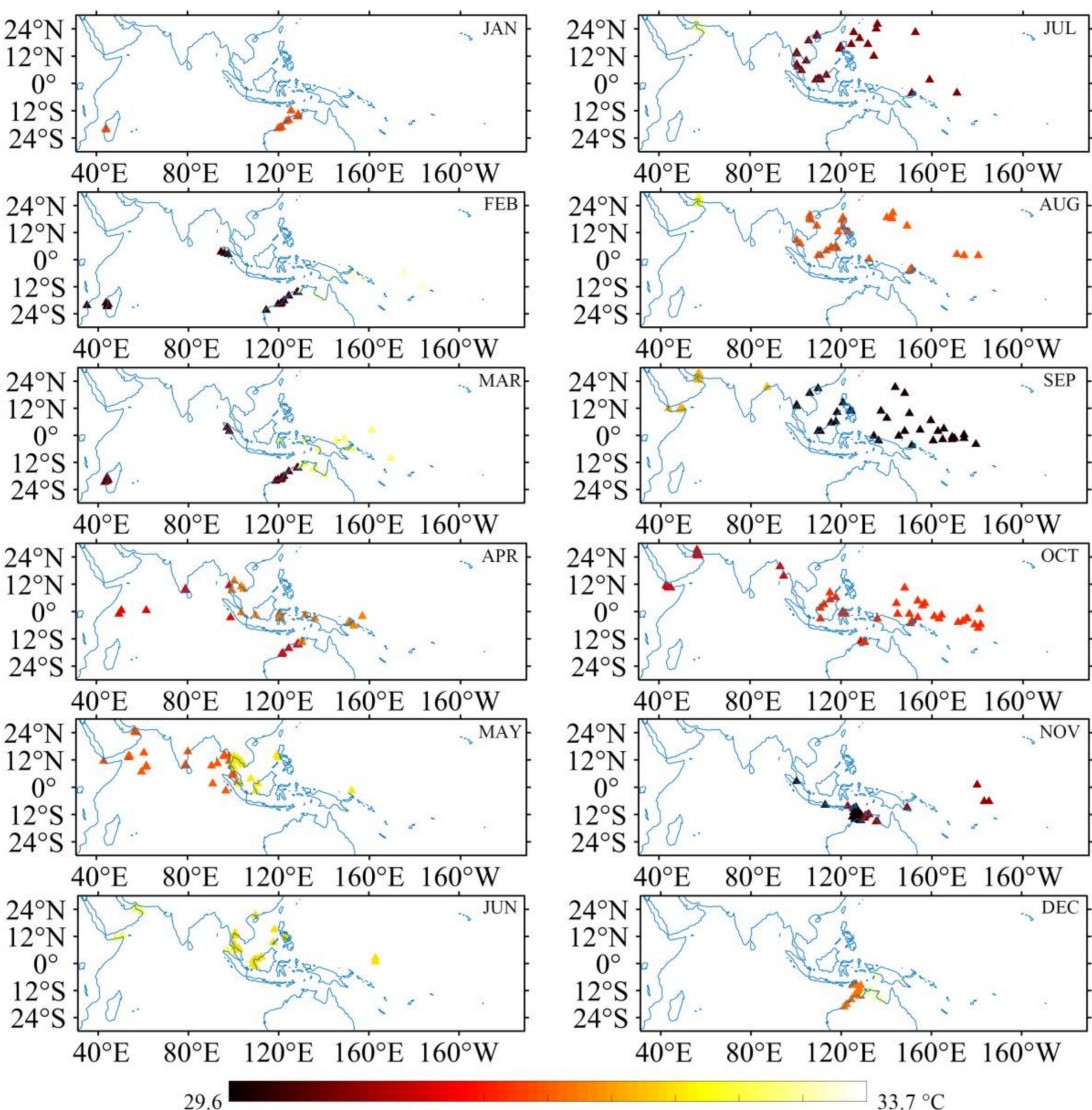

**Figure 6.** Maximum SST locations of both the IW and the PW in the calendar month.

### 3.2.2. Interannual Variabilities of the Indo-Pacific Warm Pool in Separated Demarcations

Despite the significant seasonal variations of the warm pool, long-time scale variabilities are still unneglectable. Therefore, the wavelet analysis method is utilized to perform a more detailed investigation on the warm pool's time-frequency variation patterns, for improving our understanding of the Indo-Pacific Warm Pool's multi-time scale variation regularity. The three-month running mean is applied to the time series of the warm pool properties after the monthly climatology and long-term trends are removed to obtain their interannual anomalies. The wavelet analysis results are shown in Figures 7–9, where the left panels are the amplitude spectrum and the right panels are the mean power of the warm pool properties.

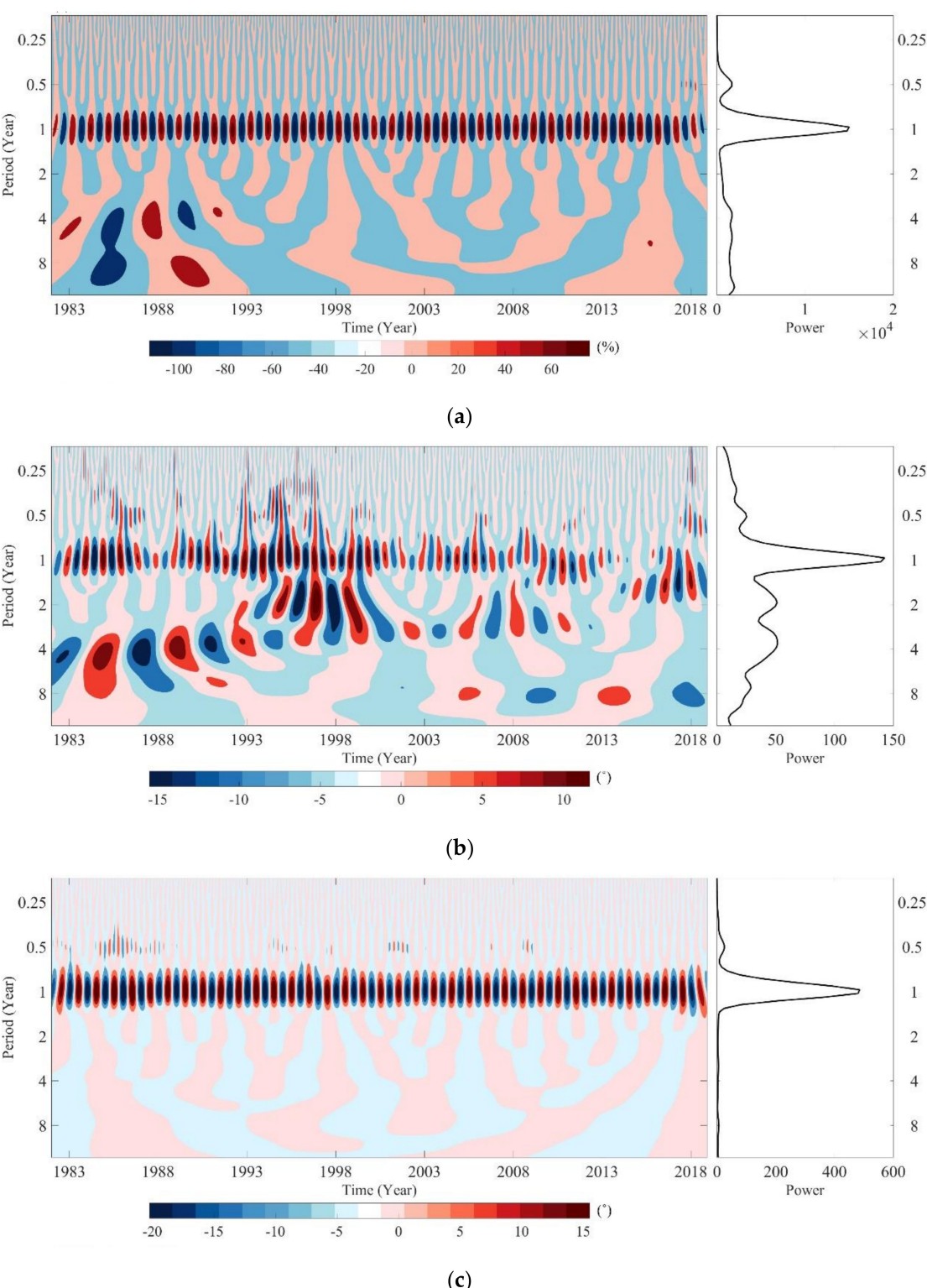

**Figure 7.** *Cont.*

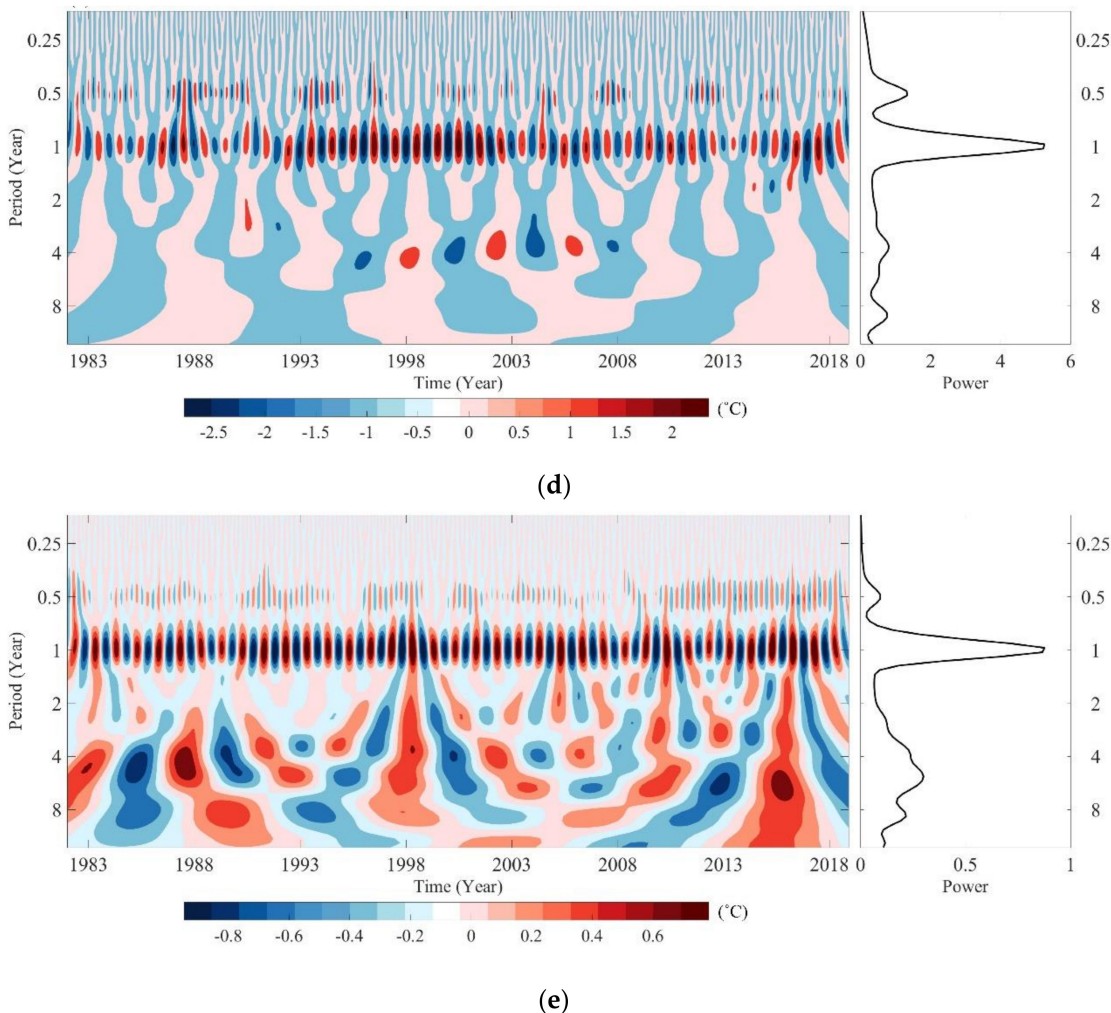

(**d**)

(**e**)

**Figure 7.** Wavelet transform spectrum and corresponding mean power of (**a**) Size variation; (**b**) Zonal center movement; (**c**) Meridional center movement; (**d**) Maximum SST; (**e**) Mean SST of the IW.

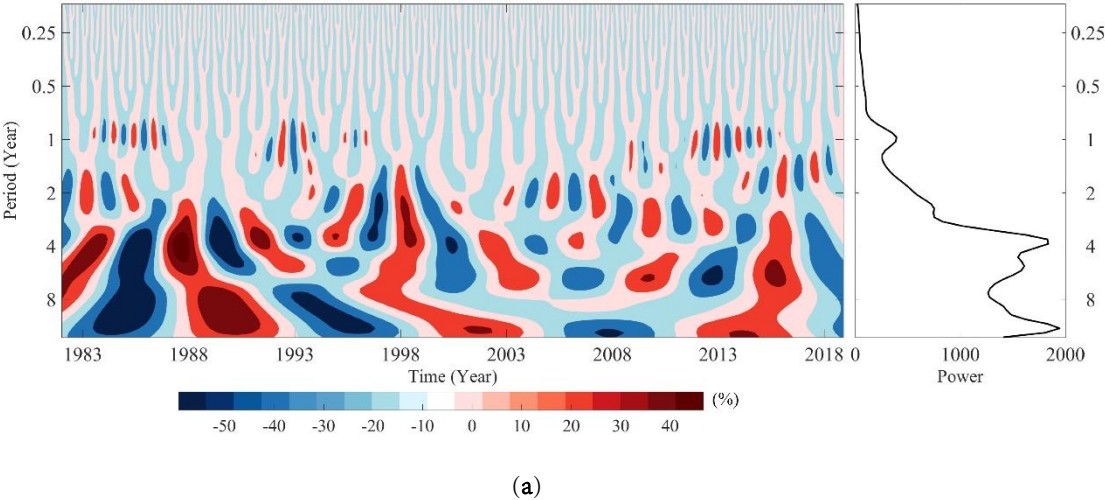

(**a**)

**Figure 8.** *Cont.*

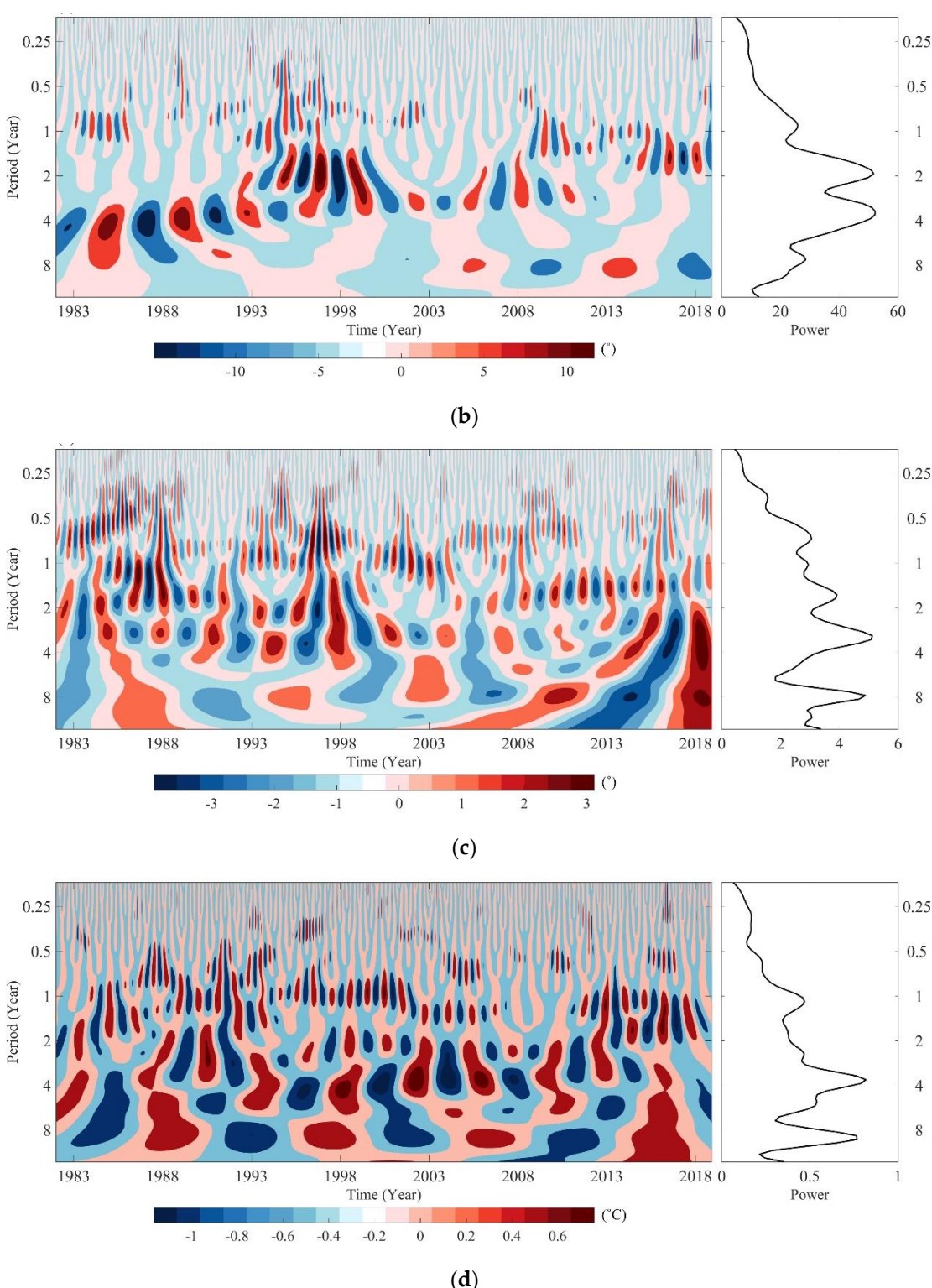

**Figure 8.** *Cont.*

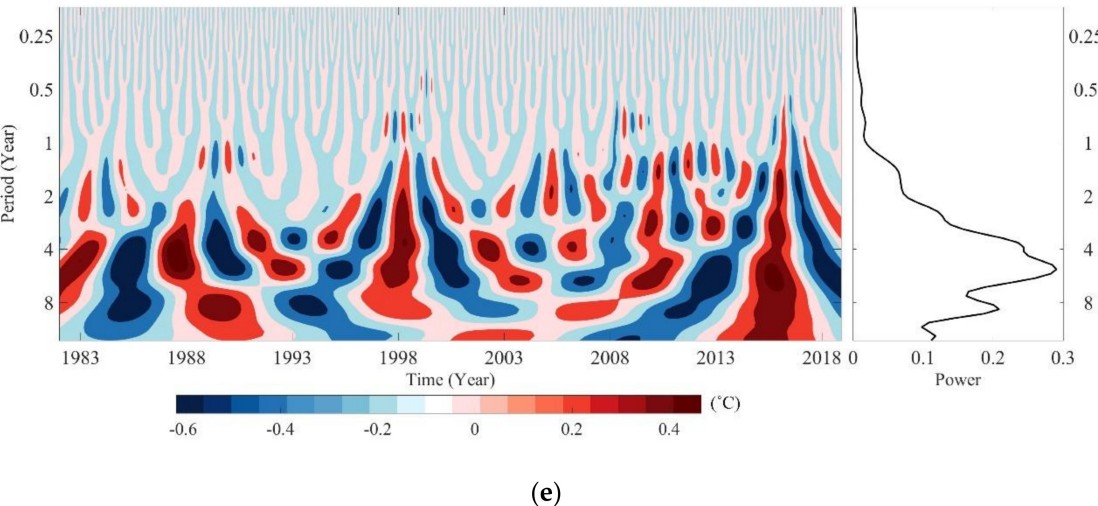

(**e**)

**Figure 8.** Wavelet transform spectrum and corresponding mean power of the seasonal cycles removed series of (**a**) Size variation; (**b**) Zonal center movement; (**c**) Meridional center movement; (**d**) Maximum SST; (**e**) Mean SST of the IW.

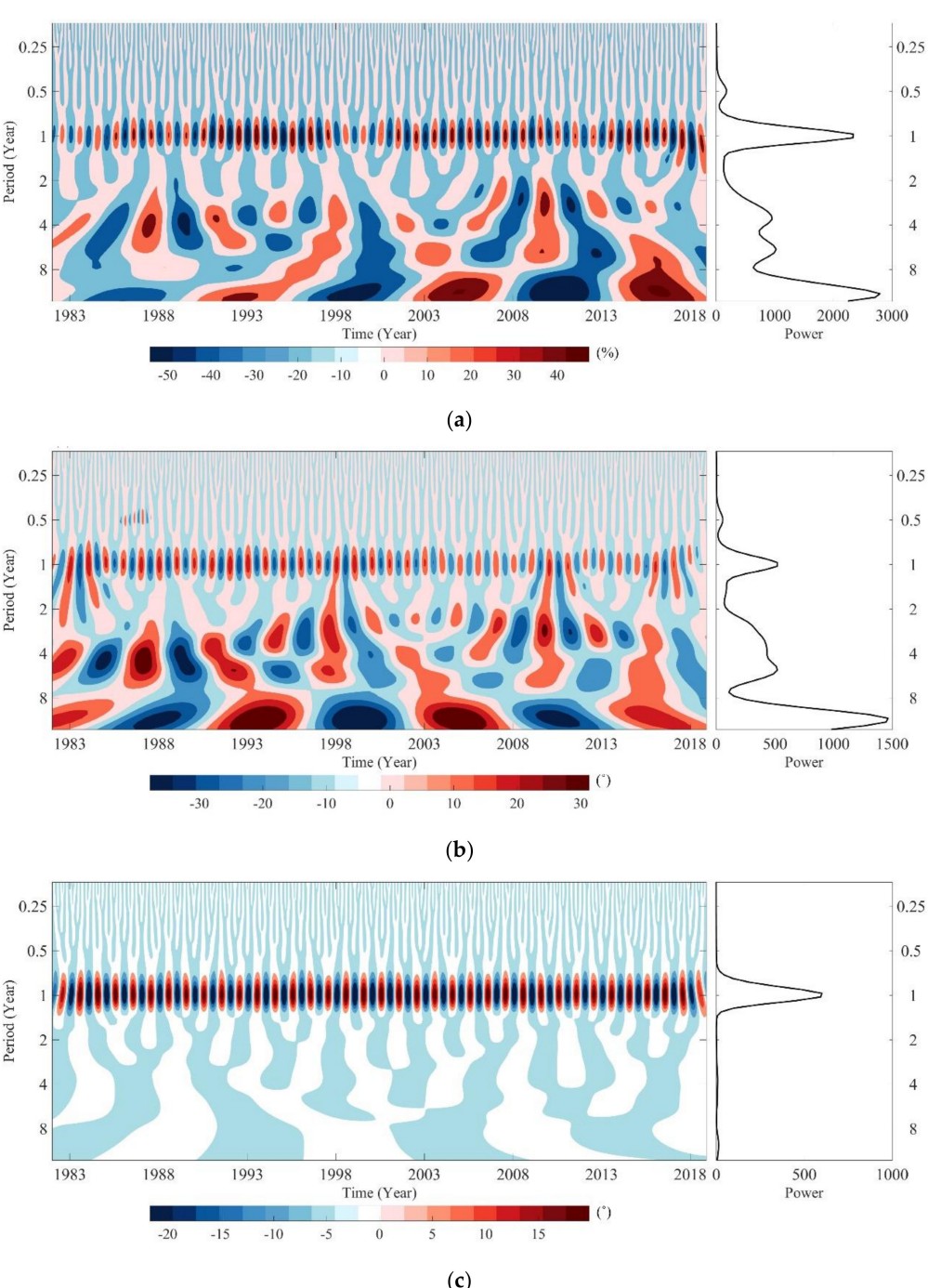

(**a**)

(**b**)

(**c**)

**Figure 9.** *Cont.*

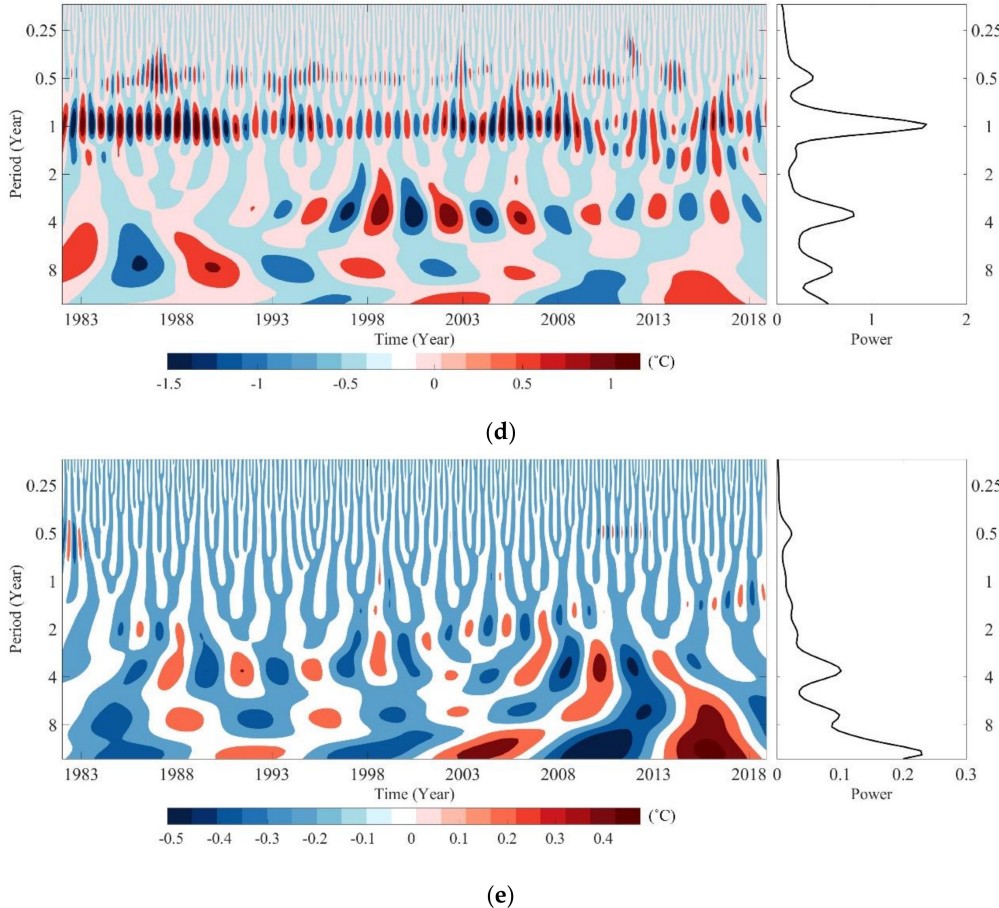

**Figure 9.** Wavelet transform spectrum and corresponding mean power of (**a**) Size variation; (**b**) Zonal center movement; (**c**) Meridional center movement; (**d**) Maximum SST; (**e**) Mean SST of the PW.

Both the amplitude spectrum panels and the mean power panels in Figure 7 show that all the properties are characterized by predominant annual cycles, which have already been proved previously. Moreover, we find that the zonal movement has evident interannual variations. The interannual variations (3-5 years) were vigorous in the 1980s and biennial variations were prominent in the mid-1990s to the early 2000s, but since the early 2000s, all the interannual signals weakened.

To discover more about the interannual variation patterns of the IW, all the annual cycles are removed before the wavelet transform is applied to the warm pool properties, and the results are illustrated in Figure 8. Although the mean power of the signals is much lower, the interannual signals can be seen clearly. Especially the interdecadal variation and the interannual variation (3-5 years) of the IW size can be seen all along the study period, and they were stronger before the mid-1990s. Concerning the meridional movement and the intensity of the IW, their variations are similar in the prominent interannual variation (3-5 years).

Figure 9 is the same as Figure 7 but for the PW. The PW has comparable interannual variations and annual variation, which are different from the IW and coincide with Figure 5. Furthermore, the size, zonal movement and mean SST of the PW have even more obvious interdecadal variations than annual variations, and the interdecadal variations have intensified since the beginning of the 21st century. The meridional center of the PW has only annual variations due to the control of the solar heating march. There are unneglectable interannual variations (3-5 years) during the mid-1990s to mid-2000s of the maximum SST. However, the mean SST barely has annual variation, which means the PW's intensity is scarcely influenced by the seasonal evolution and its movements.

The analyses presented in Figures 7–9 together indicate that the interannual variabilities of the warm pool are more vigorous in the Pacific Ocean sector than in the Indian Ocean sector, while the

seasonal variabilities are comparable in the two sectors. The meridional movement in the IW and the PW can be seen as totally controlled by the solar heating march, based on their neglectable low mean power. The size and intensity show significant interannual variations (3-5 years) in the IW, while interdecadal variations are more distinct in the PW.

### 3.3. The Relationships Between the Dominant Climate Indices and the Indo-Pacific Warm Pool

There are certain possible contributors to the variability in the Indo-Pacific Warm Pool, such as the Indian Ocean Dipole (IOD), ENSO and so on. Therefore, we firstly investigate the respective contributions of the dominant climate patterns in the Indian Ocean and the Pacific Ocean to the warm pool properties of each sector, using monthly warm pool properties and the monthly climate indices. Table 1 displays the correlation coefficients between the properties and the climate indices for boreal winter (December-January-February; DJF), spring (March-April-May; MAM), summer (June-July-August; JJA) and autumn (September-October-November; SON), respectively.

**Table 1.** The correlation coefficients between the climate indices and seasonal warm pool properties of the IW and PW.

| Warm Pool Properties | Climate Indices | IW | | | | PW | | | |
|---|---|---|---|---|---|---|---|---|---|
| | | DJF | MAM | JJA | SON | DJF | MAM | JJA | SON |
| Size | Nino 3.4 | **0.44** | **0.57** | 0.25 | 0.27 | **0.59** | **0.62** | **0.58** | **0.63** |
| | EMI | 0.07 | 0.14 | −0.31 | −0.12 | 0.28 | 0.27 | 0.25 | 0.33 |
| | IOD | 0.10 | −0.15 | 0.22 | 0.20 | −0.01 | 0.11 | 0.08 | 0.29 |
| | IOBW | **0.96** | **0.95** | **0.97** | **0.95** | **0.85** | **0.81** | **0.69** | **0.70** |
| Zonal Movement | Nino 3.4 | −0.31 | −0.06 | −0.25 | **−0.57** | **0.91** | **0.82** | **0.88** | **0.89** |
| | EMI | −0.03 | −0.10 | 0.12 | −0.24 | **0.71** | **0.68** | **0.59** | **0.79** |
| | IOD | **−0.54** | **−0.69** | **−0.86** | **−0.83** | 0.13 | 0.05 | 0.22 | **0.55** |
| | IOBW | **−0.63** | −0.16 | **−0.39** | **−0.66** | 0.30 | **0.41** | 0.30 | 0.16 |
| Meridional Movement | Nino 3.4 | **0.46** | **−0.58** | −0.13 | 0.29 | −0.32 | **−0.35** | **−0.42** | **−0.50** |
| | EMI | **0.38** | 0.03 | 0.10 | 0.16 | −0.21 | **−0.39** | **−0.62** | **−0.65** |
| | IOD | **0.35** | **0.48** | 0.04 | **0.49** | −0.06 | −0.22 | 0.10 | −0.23 |
| | IOBW | **0.59** | **−0.55** | −0.20 | −0.05 | 0.15 | −0.26 | 0.09 | 0.17 |
| Maximum SST | Nino 3.4 | 0.33 | **0.55** | 0.08 | 0.12 | 0.17 | **0.53** | −0.10 | −0.16 |
| | EMI | 0.11 | 0.17 | −0.17 | −0.33 | −0.10 | 0.08 | −0.27 | **−0.37** |
| | OD | −0.31 | −0.11 | −0.01 | 0.05 | −0.17 | **−0.37** | −0.29 | −0.19 |
| | IOBW | **0.61** | **0.82** | 0.27 | **0.35** | 0.13 | **0.58** | **0.42** | **0.41** |
| Mean SST | Nino 3.4 | **0.61** | **0.67** | 0.31 | **0.37** | **0.44** | **0.46** | 0.00 | 0.13 |
| | EMI | 0.19 | 0.19 | −0.31 | −0.14 | 0.25 | **0.35** | −0.09 | −0.09 |
| | IOD | 0.00 | −0.06 | 0.18 | 0.28 | −0.01 | 0.21 | −0.05 | −0.01 |
| | IOBW | **0.95** | **0.99** | **0.95** | **0.94** | **0.72** | **0.79** | **0.63** | **0.69** |

The bold values are statistically significant above the 95% level.

As shown in Table 1, the size and mean SST of each sector are highly correlated with IOBW in every season, which means the warm pool always covaries with basin-wide warming or cooling. The correlations between Niño 3.4 and the PW size and zonal movement are significant and persistent throughout the ENSO's lifecycle. When El Niño events occur, the PW expands as the zonal center shifts eastward and the meridional center shifts southward (vice versa for La Niña events). The PW Mean SST significantly correlates to Nino 3.4 in boreal winter and spring, when ENSO is on the peak and decay stages, indicating that El Niño events can enhance the PW intensity. As for the IW, it is found that ENSO has a remote influence on the size, mean SST and meridional movement in boreal winter and spring. Since ENSO typically peaks in winter, the significant correlations indicate that when El Niño (La Nina) events break out, the IW would expand (shrink) with its meridional center moving northward (southward) and the mean SST of it would increase (decrease).

Unsurprisingly, the zonal centers of the IW and the PW are significantly related to the climate modes belonging to their ocean, respectively. The PW zonal center moves eastward(westward) when the Pacific Ocean is on the ENSO positive(negative) phase, whether it is EP mode or CP mode. Furthermore, the EP ENSO has a stronger correlation with the zonal movement. The PW size is strongly

related to Nino 3.4, while the significant correlation only occurs in winter and spring between the mean SST and Nino 3.4. The IW zonal center moves westward(eastward) when there is a positive(negative) IOD phase or IOBW phase. Interestingly, the size and mean SST of the IW are strongly correlated with Nino 3.4 in winter and spring, which is on the peak and the decay stage of ENSO events.

On account of the enormous correlations between IOBW and the properties of IW and PW, to further reveal the cause-and-effect relations between the leading climate mode and warm pool variations, their lead-lag correlations are shown in Figure 10. The IOBW shows the most significant correlations with the IW size (R = 0.85), the IW zonal center (R = 0.34) and the IW mean SST (R = 0.89) without any lag. The IW maximum SST (R = 0.32) lags to the IOBW for one month when the largest correlation occurs. Meanwhile, the meridional center movement is not strongly linked with the IOBW. The largest correlations between the IOBW and the PW are the time when the PW size (R = 0.51) leads the IOBW by two months, the PW zonal center (R = 0.50) leads the IOBW by three months, the maximum SST (R = 0.32) lags the IOBW by two months and the mean SST (R = 0.42) lags the IOBW by one month.

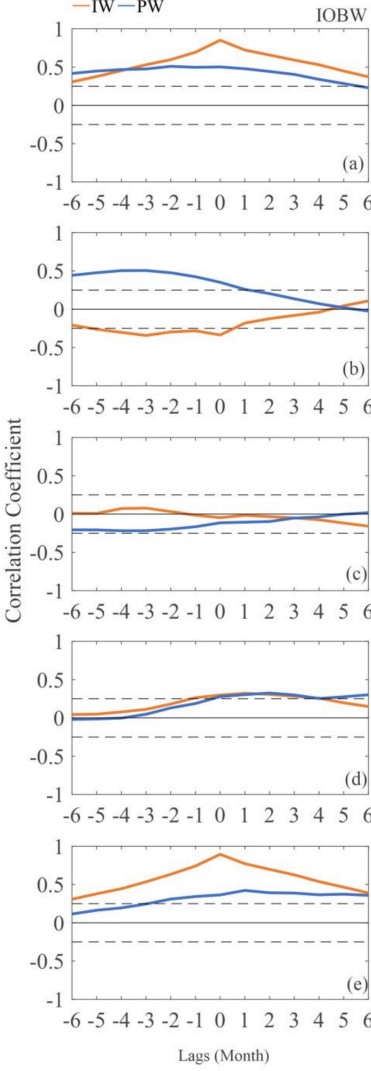

**Figure 10.** Lead-lag correlations between IOBW and the (**a**) Size variation; (**b**) Zonal center movement; (**c**) Meridional center movement; (**d**) Maximum SST; (**e**) Mean SST of the IW and PW.

## 4. Discussion

Using the long time series SST dataset, we reveal seasonal and interannual variation characteristics of the Indo-Pacific Warm Pool through the study period (1982-2018). By separating the IPWP into the Indian Ocean Sector and the Pacific Ocean Sector, the respective variation characteristics and their associations with different climate modes are demonstrated.

The IW is not as large as the PW; when it occupies most of the IPWP in March, it only accounts for 44.27%. Whereas the PW is 75.59% of the IPWP in August when it occupies the IPWP the most. The seasonal variabilities of the IW and PW turn out to be very different. First of all, the season evolution of the IPWP size and mean SST is dominated by the vigorous seasonal variation of the IW. In the following, the seasonal variation patterns are different. The IW's expansion is accompanied by a westward shift and an increase in mean SST, and vice versa. Considering the above three properties, the IW reaches its peak status in late boreal spring and its weakest status in autumn. The PW size, zonal and meridional movements and maximum SST are highly correlated. The PW expands to its largest together with moving to westernmost and northernmost in late boreal summer. Oppositely, the PW shrinks to its smallest when it moves to its easternmost and southernmost in boreal winter. The conclusions indicate that the IW varies along with the Indian monsoon, while the PW accompanies the seasonal solar heating march. Specifically, the warm pool in both sectors moves meridionally with the moving of the sun's incident point, but the movement has little influence on the IW size or intensity. In particular, the noteworthy zonal movement of the IW is that the direction turns over in boreal summer due to the north boundary of the Indian Ocean uniquely constituted by the Asian continents as well as the southwesterly Indian monsoon during boreal summer inducing coastal upwelling and enhancing surface evaporation cooling in the western Arabian Sea. Moreover, it turns out the IW mean SST is more accurate to represent the intensity. This is because the IW maximum SSTs locate off the coast of Australia or Madagascar strait in austral spring and summer and locate in the Gulf of Oman or coast of the Arabian sea or Bay of Bengal in boreal spring and summer. Meanwhile, the mean SST has a clearer seasonal cycle. Oppositely, the maximum SST is more suitable for representing the PW intensity. Because the PW mean SST is only regulated by the sun's incident point, which means only when the sun shines on the tropic of Cancer or the tropic Capricorn, the mean SST is the highest. However, the maximum SSTs' locations are not confined by the continents as the IW maximum SSTs. It has an evident seasonal cycle in which it moves from the north coast of Australia, then to the Maritime Continent, then to the northwest Pacific. Together with the movement, the maximum SST decreases. The IW has weaker interannual variability compared to its seasonal variability, which is comparable with the PW interannual variability. The wavelet transformed results indicate that the IW has an enormous annual signal and relatively weaker interannual variation signals (3–5 years), but the PW has comparable signals of annual and interdecadal. Except for the meridional movements, all the other properties of the same basin have similar interannual variation signals.

Correlation coefficients of the warm pool and the dominant climate modes indicate that the IW and PW are significantly correlated with the basin-wide warming or cooling with regards to their size and mean SST. CP-ENSO is also highly correlated with the warm pool of the two sectors, which can be depicted as when there are El Niño events both sectors are expanding, with the PW moving northward and eastward but the IW moving southward and westward. However, more sophisticated experiments and simulations to explore and clarify the mechanisms of the warm pool variability in future work are still needed.

## 5. Conclusions

In this study, five warm pool properties (surface size, zonal center, meridional center, maximum SST and mean SST) were applied to examine the variabilities of the IPWP during 1982-2018 and to contrast the characteristics of the Indian Ocean sector and Pacific Ocean sector. We utilize harmonic analysis and normalized series of warm pool properties to explore the seasonal variabilities of the properties. We find that on the seasonal time scale, the oscillation of the IW is much stronger than

the PW on size and intensity. However, on the long-term time scales, variabilities of the IW and the PW are comparable. The IW size is more related to the zonal movement and the IW mean SST can be the more precise indicator of the intensity on the seasonal scale. The PW size is related both to zonal and meridional movements, and the maximum SST is the accurate indicator of the intensity on the seasonal scale. Furthermore, the wavelet analysis is utilized to depict the time-frequency features of the properties of the IW and PW. It turns out the IW has more significant interannual variations (3-5 years), while the PW has interdecadal variations. Except for the meridional movements, all the other properties of the same basin have similar interannual variation signals. We also analyzed the correlations between the climate indices and the warm pool properties for a better understanding of the possible physical mechanisms. According to the correlation analysis, basin-wide warming or cooling is found to have the strongest interactions with the IW and the PW. The differences and similarities of the variabilities of the IW and PW and their major forcing features found in this study could help improve the understanding of the Indo-Pacific Warm Pool with its associated climate changes on marine and atmospheric circulation.

**Author Contributions:** Conceptualization, Q.D.; Data curation, F.K.; Formal analysis, Z.Y.; Investigation, Z.Y.; Methodology, S.L.; Supervision, Q.D.; Writing – original draft, Z.Y.; Writing – review & editing, F.K. and D.C. All authors have read and agreed to the published version of the manuscript.

**Funding:** This research was supported by the National Natural Science Foundation of China (NO. 41876210), and the National Key Research and Development Program of China (2017YFA0603003).

**Acknowledgments:** The SST data is from the NOAA/OAR/ESRL PSD (www.esrl.noaa.gov/psd). We are grateful to the NOAA for access to the data.

**Conflicts of Interest:** The authors declare no conflict of interest.

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
