# Peer review of "Seasonal and Interannual Variability of the Indo-Pacific Warm Pool and its Associated Climate Factors Based on Remote Sensing"

_remotesensing, doi:10.3390/rs12071062_

Round 1

Reviewer 1 Report

The manuscript “Seasonal and Interannual Variability of the Indo-Pacific Warm Pool and its Associated Climate Factors based on Remote Sensing” is to some extent repetition of the work of

Kim et al., The distinct behaviours of Pacific and Indian Ocean warm pool properties on seasonal and interannual time scales, JGR, 117, D05128,  2012.

Unlike the cited work, the authors have based their study on the analysis of the sea surface temperature (SST) from satellite measurements. The presentation and interpretation of the results, need, however, substantial improvements before the paper will become suitable for publication. 

More details about reviewer's questions, comments and recommendations are given in attached file.

Reviewer 2 Report

This study analyzed the variability of the Indian Ocean warm pool and the Pacific Ocean warm pool using the SST data with the wavelet analysis method.

Their main conclusions are characterizations of the two warm pools by correlating with other climate indices, such as ENSO and IOD.

The overall method and conclusions are well described with details. However, I have one concern is the lack of a satisfied explanation of the underline physics. For example, what is the physical/dynamic reason for the variability or the concluded characters of the two warm pool? By the way, a simple correlation between the warm pools with the ENSO/IOD is not sufficient enough to prove their casual relationships.

One suggestion is using a coupled ocean-atmosphere model to investigate the physical/dynamic reasons--experiments to test a hypothesis by removing one of the possible factors.

Author Response

Point 1: This study analyzed the variability of the Indian Ocean warm pool and the Pacific Ocean warm pool using the SST data with the wavelet analysis method.

Their main conclusions are characterizations of the two warm pools by correlating with other climate indices, such as ENSO and IOD.

The overall method and conclusions are well described with details. However, I have one concern is the lack of a satisfied explanation of the underline physics. For example, what is the physical/dynamic reason for the variability or the concluded characters of the two warm pool? By the way, a simple correlation between the warm pools with the ENSO/IOD is not sufficient enough to prove their casual relationships.

One suggestion is using a coupled ocean-atmosphere model to investigate the physical/dynamic reasons--experiments to test a hypothesis by removing one of the possible factors.

Response 1: We would like to sincerely thank the reviewer for the valuable suggestion. In this revision, we focused on explaining the variabilities more specifically. It is a pity that we did not carry out a simulation experiment, but it will be a goal for us to use it to investigate the forcing factors and physical mechanisms.

Special thanks to you for your valuable suggestions for our work.

Round 2

Reviewer 1 Report

The corrected manuscript is much concise and easier understandable. There are still some minor omissions, which should be properly addressed:

  • Some unclear mixture is appeared in Fig.4 - some of panels are doubled, which has to be corrected.
  • The authors’ response to the reviewer comment to Fig.9 is that they remove the correlations of IW and PW with Nino.4, IOD and EMI, but the old version of Fig.9 still exists in the manuscripts and should be removed.

I would like to suggest to the authors to keep in mind my questions, and to try to find proper answers in their further research.

Author Response

Representing my cowriters, I am deeply grateful for your comments and suggestions. Your comments are valuable, and your suggestions mean a lot to not only this research but also in my further research. We will keep your suggestions in mind to always try to find the proper answer and to express our ideas more specifically.

Point 1: Some unclear mixture is appeared in Fig.4 - some of panels are doubled, which has to be corrected.

Response 1: We are sorry for our negligence in the figure’s disorder, and we have revised it in the manuscript.

Point 2: The authors’ response to the reviewer comment to Fig.9 is that they remove the correlations of IW and PW with Nino.4, IOD and EMI, but the old version of Fig.9 still exists in the manuscripts and should be removed.

Response 2: We are sorry for our negligence in the figure’s disorder, and we have deleted the needless panels in the manuscript.

We also rearranged the abstract, discussion and conclusion part. We appreciate the reviewers’ warm work earnestly and hope that the correction will meet with approval.

Once again, thank you very much for your comments and suggestions.

Reviewer 2 Report

This is my second time review this paper: The revised version has addressed my concerns.